# Turning to ‘Trusted Others’: A Narrative Review of Providing Social Support to First Responders

**DOI:** 10.3390/ijerph192416492

**Published:** 2022-12-08

**Authors:** Anna Tjin, Angeline Traynor, Brian Doyle, Claire Mulhall, Walter Eppich, Michelle O’Toole

**Affiliations:** RCSI University of Medicine and Health Sciences, D02 YN77 Dublin, Ireland

**Keywords:** first responder, social support, mental health, post-traumatic stress, trusted other

## Abstract

First responders, such as paramedics and firefighters, encounter duty-related traumatic exposures, which can lead to post-traumatic stress (PTS). Although social support protects against PTS, we know little about how first responders’ families, spouses/partners, friends, and care-partners (i.e., ‘trusted others’) provide social support. This narrative review explores support behaviors, coping strategies, and resources trusted others use to support first responders. A structured literature search yielded 24 articles. We used House’s (1981) conceptual framework to inform our analysis. We identified three main themes: providing support, finding support, and support needs. Additionally, we describe trusted others’ self-reported preparedness, coping strategies, and barriers to providing social support. We found that trusted others provided different types of support: (a) emotional (fostering a safe space, giving autonomy over recovery, facilitating coping mechanisms, prioritizing first responders’ emotional needs); (b) instrumental (prioritizing first responders’ practical needs, handling household tasks, supporting recovery); (c) appraisal (active monitoring, verbal reassurance, positive reframing), and (d) informational (seeking informal learning). In their role, trusted others sought formal (organizational) and informal (peer and personal) support and resources, alongside intrapersonal and interpersonal coping strategies. Identified barriers include inadequate communication skills, maladaptive coping, and disempowering beliefs. Thus, we offer practical, treatment, and social support recommendations.

## 1. Introduction

First responders (FRs) comprise all specialized professionals trained to assist at the scene of an emergency [1], often risking their lives in service to their communities [1,2]. They include military, emergency medical service workers (i.e., paramedics), law enforcement officers, and firefighters [3] who face complex professional and interpersonal challenges. These occupational challenges often translate to chronic physical and psychological health conditions [4], such as respiratory issues [2], a higher risk of cardiovascular disease [5], and trauma-related stress and disorders [5,6,7,8]. However, the cumulative impacts of duty-related traumatic exposures can result in either post-traumatic stress (PTS) or post-traumatic growth (PTG). PTS is a common adaptive reaction to traumatic events or incidents that subside after days or weeks [9]. On the contrary, post-traumatic stress disorder (PTSD) is a clinically diagnosed condition listed in the 5th edition of the Diagnostic and Statistical Manual of Disorders (DSM-5) as a mental illness diagnosis [9]. PTS could lead to the development of post-traumatic stress disorder (PTSD) through the overwhelming accumulation of stressors exceeding individuals’ ability to cope and process [9]. Alternatively, these events can also have a positive psychological impact on first responders. The positive PTG response to traumatic incidents can enhance first responders’ resilience, professional self-efficacy, and perception of their own lives [10]. Critical Incident Stress Management (CISM) interventions have been widely established across many organizations that employ emergency service personnel to limit the negative impact of critical incidents [8]. Unfortunately, many first responders do not avail of organizational support, instead turning to informal support from family, friends, and care-partners (i.e. trusted others). We are only beginning to delineate the scope, scale, and potential of trusted others’ support in the health and wellbeing of first responders. In addition, these family members and close friends also experience hardship in providing support for first responders [11].

Evans et al. [11] defined trusted others as informal care-partners of individuals with life-limiting conditions. We apply the term ‘trusted others’ in this context, referring to an inclusive construct describing first responders’ informal social support networks comprising spouses or partners, family members, care-partners, and close friends. Multiple studies capture the impact of secondary trauma exposures on trusted others, with first responders frequently bringing home this occupational baggage [12,13]. Figley [13] defined secondary trauma as stress experienced by someone due to helping others who are suffering or traumatized. Fortunately, trusted others also reported positive effects, such as a sense of pride and loyalty in the work of first responders [14,15]. Thus, trusted others represent a potentially essential source of social support for first responders.

Social support is the “emotional and practical assistance that one receives from their social groups, such as family or friends, and colleagues, in times of crisis and distress” [16] (p. 2253). Social support mitigates PTS associated with critical incident exposure and enhances wellbeing and PTG [17,18,19,20]. Early social support strengthens this protective impact, reducing the need for further professional psychological interventions [17]. In emergency services, the most prevalent form of early intervention, known as ‘psychological first aid,’ is traditionally provided by colleagues, with more recent research suggesting family members and close friends as a practical alternative [21]. Preferences for early intervention support differ, with both professional rank and duration of service influencing the type of desired support [22,23]. For example, firefighters prefer to talk to spouses, family members, or friends [22], while law enforcement officers and those with more service years opt to speak to professionals about their work-related stressors [22]. House [24] characterizes social support as the perception and reality that someone feels cared for and has access to assistance from others. Strong social support networks naturally encourage help-seeking from others. There are four types of support [24]:Emotional support refers to providing trust, empathy, love, and care for the person seeking help.Instrumental support offers tangible services or practical help.Appraisal support includes providing affirmation or information relevant for self-evaluation.Informational support provides knowledge.

Trusted others’ readiness represents a state of willingness or receptivity to an experience or activity or a willingness to act [14,15] that also requires a sense of preparedness. Although we know that friends and family are willing to support their first responders [25], we do not yet understand sufficiently how prepared they are to do so. Preparedness is being able, ready, and acting in a given capacity [26]. For trusted others, formal supports to enhance perceived readiness include communicative, social, or therapeutic experiences with healthcare professionals or structured organizational support to minimize the impact of secondary stress. Informal support encompasses all other efforts without the help of a mental health professional or organization-based structural support that trusted others engage in to alleviate the effects of secondary stress (e.g., CISM services). However, many questions remain. What factors shape capacity for trusted others willing to engage as support partners? Do trusted others avail of the support and resources they need to prepare for their role? What formal and informal resources and strategies do they use? Given the significant burden on first responder families and the demonstrated long-term impacts, we must expand the research base on the preparedness of trusted others to provide social support to maximize this vital mechanism to enhance first responder health and wellbeing.

This narrative review aims to summarize current understanding, identify unknowns, and ‘map the gaps’ [27] to promote a research agenda in this area. To achieve our aim, we used these guiding research questions: How do trusted others currently provide social support to their first responders?How are trusted others prepared to support their first responders (their readiness, coping strategies, and available resources)?How can trusted others be best prepared to provide social support?

## 2. Materials and Methods

A preliminary review of the literature revealed the limited and unfocused nature of empirical studies on the social support provided by first responders’ trusted others. We adopted a narrative review methodology [28], given its exploratory nature and significant gaps in the literature. Narrative reviews offer a transparent approach to summarizing and exploring diverse subject matter by explaining and interpreting evidence while incorporating various research methods [29]. This methodological approach is rigorous, flexible, and transparent and can produce a thoughtful, practical synthesis of the topic [28].

In consultation with a medical reference librarian, we developed a comprehensive search strategy, without date limiters, to identify potentially relevant articles using Medline, EMBASE, ERIC, PsycINFO, Web of Science, Scopus, and Google Scholar using a documented search strategy (Table 1). We searched for studies using qualitative, quantitative, or mixed methodologies, book chapters, literature reviews, thesis dissertations, and grey literature published in English. Review articles were utilized to identify additional literature and were excluded from data extraction.

### 2.1. Inclusion and Exclusion Criteria

We included articles that described social support provided by family members, partners, spouses, friends, or care-partners, narratives surrounding support and coping strategies, resource preference, availability, and barriers experienced during social support provision (Table 1). We defined first responders as those employed predominantly in public service, including military personnel and veterans, law enforcement officers, firefighters, paramedics, or Emergency Medical Technicians (EMTs) [30,31]. Frontline healthcare professionals were included due to their similar occupational risks and the lack of documented social support [2,31]. As our study focused on first responders’ trusted others, we excluded those outside the scope of our definition of trusted others, as well as non-first responders. During the review of the literature, we found numerous studies focusing on first responders with little acknowledgment of the trusted others. To best capture support provision and needs of trusted others, we excluded studies that did not acknowledge or describe trusted others’ involvement in supporting first responders, their resource utilization, and barriers experienced. We have also excluded non-English papers and papers with no full-text available.

### 2.2. Data Extraction and Synthesis

The search strategy yielded 125 potential references. The authors screened abstracts and titles for inclusion criteria and reviewed the remaining 24 full-text articles and abstracted data, including study referencing, sample characteristics, and study design. To support full-text review, a standardized data extraction form was developed in MS Excel, with MS Word used for the analysis and synthesis of findings.

Preliminary findings were coded, analyzed, and interpreted using iterative categorization (IC) techniques that allow text coding according to topic, concept, or theme [32]. IC was chosen due to its rigor and simplicity, allowing for an initial deductive coding using House’s [24] conceptual framework, followed by inductive coding to analyze further the themes we initially identified with thematic analysis [33,34]. The authors reviewed the summaries of the final 24 papers, and an initial coding scheme was compiled with definitions comprising 43 codes with quotes or definitions supporting each. These codes were categorized according to the three research questions. Each article was then coded, discussed, and rearranged or collapsed accordingly. After consultation with the wider team, this resulted in three core themes and eight sub-themes.

### 2.3. Reflexivity

We considered the positionality of the research team throughout, namely that of “outsider researchers” [35]. Our team included a comprehensive combination of retired first responders, mental health researchers, and educators.

Two authors are retired operational firefighters/paramedics with a deep understanding of both the context and culture of first responder professions (M.O’T. and B.D.). A.Tj. has a research background in health promotion and mental health. A.Tr. is a psychologist with a PhD in Psychology and Health and a research background in mental health. C.M. is an educationalist with a PhD in Geology. W.E. is a pediatric emergency medicine physician and PhD educationalist with a history of interactions with first responders. The authors discussed their preconceptions and how these might influence the current interpretation of findings and returned to this discussion throughout the review process. The whole team discussed the final analysis and presentation of findings.

## 3. Results

Table 2 provides an overview of all included papers and demonstrates the diversity of study populations and countries of origin. Trusted others captured through the studies include the spouse or long-term partner, family members (parents and siblings), children, close friends and care-partners of law enforcement, military and veterans, firefighters, and frontline healthcare professionals (paramedics, emergency medical technicians, doctors, and physiotherapists). Literature ranges from 1995 to 2022, with over two-thirds (N = 19) of studies originating from North America and almost half (N = 11) adopting the qualitative methodology.

### 3.1. Themes and Interpretation

Our analysis and synthesis of included studies yielded rich evidence for the vital role that trusted others play in the wellbeing of their first responders in helping to mitigate the negative implications of duty-related traumatic exposures. Our main findings revealed a fundamental tension for trusted others: not only did they provide various forms of support to their first responders, they also required support themselves to engage in this at times demanding aspect of their lives. Thus, we identified three main themes related to trusted others and social support, including (a) providing support, (b) finding support, and (c) support needed by trusted others themselves. We now present our main themes and subthemes (see Table 3 for an overview) and then offer recommendations identified by trusted others for their ongoing support role. To add richness to our findings, we provide direct quotations from the original studies that exemplify each identified theme.

### 3.2. Theme 1-Providing Support

All 24 manuscripts included common behaviors among trusted others providing support to a first responder. Figure 1 provides an overview of these key findings and categorizes trusted others’ support behaviors according to House’s [24] operational definitions of support behavior. See Table 4 for definitions of these types of social support.

#### 3.2.1. Emotional Support


*“So, actually listening to what the issue is, and then being able to draw out what the actual trigger is that’s causing the stress.”*
[42] (p. 159)

Listening and providing emotional support featured prominently. Trusted others took pride in being the person first responders trusted to confide in daily [43] and when facing adversity [40,47]. Emotional support was the main pillar of support trusted others provided to their first responders (19/24 studies). These studies suggested that fostering a psychologically ‘safe space’ was central to providing emotional support for first responders [12,38,39,40,41,42,52,53,56]. This included being present by providing time and emotional capacity [37], being a ‘sounding board’ [40,52], listening actively [42,53,56], expressing acceptance and open-mindedness [39], observing and recognizing first responders’ emotional condition [41], and giving first responders space and time to decompress before trusted others’ support provision [42]. Some trusted others responded to their first responders’ distress by allowing first responders to dictate the timing and amount of support they were willing to receive [42,53].

Reconnecting with first responders by enjoying one-on-one quality time was one of several strategies trusted others used to strengthen their relationship and communication with first responders [42]. Waddell et al. [57] documented that some emotional strategies employed by trusted others include encouraging first responders’ positive coping mechanisms. Facilitating positive coping strategies, such as seeking professional help [57], practicing religious beliefs [36], and having conversations to diffuse first responders’ negative emotions [45], also allowed first responders the flexibility to process their emotions and reactions in the most effective way [45]. Bochantin [39] captured trusted others’ perceptions of first responders’ occupational risk and validation of first responders’ emotional needs. Trusted others often prioritized first responders’ emotional needs over their personal feelings by concealing their fear over first responders’ safety [39] and providing emotional support despite their emotional condition [42].

#### 3.2.2. Instrumental Support


*“It was a war; we were on the home front holding it down.”*
[51] (p. 910)

Instrumental or practical support was reported predominantly by female partners of first responders [12,47,51]. Spouses and partners handled main household tasks and took on additional domestic responsibilities to reduce strain on first responders [42,47]. Trusted others indicated that being flexible and accommodating toward first responders’ schedules was a significant part of their family life [12,45,56], where their work shift significantly influenced childcare, mealtimes, and special events. In several studies involving veterans living with PTSD [11,40,57], trusted others played a crucial part in their treatment plan. First responders relied on trusted others to arrange medical appointments or therapy sessions and advocate their needs to healthcare professionals. Trusted others reported increasing time and capacity to care by sacrificing personal time [37].

#### 3.2.3. Appraisal Support


*“Jane reported that she had to learn to be different. To read his moods and know when he has had a bad day and try to encourage him.”*
[52] (p. 48)

Trusted others typically offered appraisal support to their first responders’ experience following adverse events [40,45] or when they showed signs of PTSD that impacted others [11,51]. Evans et al. [11] identified how trusted others actively monitored first responders’ triggers and symptoms to understand their needs and provide feedback for them in therapy sessions [11]. Trusted others mentioned that with time and experience, they learned to ‘read’ first responders and were able to determine if they should express their concern over their wellbeing [45]. Porter and Henriksen [52] captured the strategy trusted others use to give positive affirmation and verbal reassurance about first responders’ capability and motivations to work as first responders. Trusted others expressed that constant communication and affirmation were crucial to their relationship with first responders [52]. Moreover, trusted others provided appraisal support by reframing difficulties faced by first responders in a positive light [37,47]. Trusted others used the collective pronouns “we” and “us” to describe the family unit, to share the blame, and diffuse tension in the conversation [37]. Landers et al. [47] described having a “well of good times,” a collection of positive memories, as a source of resiliency in which trusted others could remind first responders of good times they experienced when faced with adversity.

#### 3.2.4. Informational Support


*“Most of what we know [about PTSD] comes from news broadcasts, the movies, or the Internet.”*
[40] (p. 746)

Lastly, trusted others provided informational support by gathering information and educating themselves about PTSD, including first responders’ treatment plans, symptoms, and triggers [40,42]. As Beks [37] reported, the prospect of recovering both their first responders and their relationship motivated trusted others to seek answers in order to understand PTSD better and facilitate healing.

### 3.3. Theme 2-Finding Support

Thirteen of the included papers reported coping strategies and the use of resources by trusted others in preparation for the role of support partner. Coping strategies and the use of resources were categorized as (a) formal and (b) informal. Figure 2 illustrates the categorization of supports available to trusted others.

#### 3.3.1. Formal Coping and Use of Resources

Formal coping and use of resources includes communicative, psychosocial, or therapeutic experiences with healthcare professionals or organizational groups to minimize the impact of secondary stress. Six papers described formal coping and the use of resources by trusted others. These included four organization-based support programs and two descriptions of support from a mental health professional. We identified three sub-themes that highlight important formal resources for trusted others: (1) mental health professionals; (2) organizational support; (3) psychoeducation, resilience, and emotional management.

Mental health professionals, including therapists and counselors, gave trusted others an outlet to find support and encouragement [42,51]. Two included papers referred to support-seeking behavior by trusted others from a mental health professional [42,51]. Spouses of first responders recognized the benefit of couples counseling in learning communication skills and understanding first responders better [42]. They also emphasize their preference for employee assistance programs when seeking counseling for themselves and their first responder. Furthermore, Menendez et al. [51] recommended therapy for first responders’ children following adverse events, emphasizing the impact on the entire family unit.

All four organization-based support programs that included trusted others directly were USA military or veteran based. These included the Warrior Resilience and Thriving (WRT) program [46], the Coming Home Project (CHP) [38], the Families Over Coming Under Stress (FOCUS) Family Resilience Program [48,54], and the Veterans Affairs Program of Comprehensive Assistance for Family Caregivers (PCAFC) [55]. These organized, formal sources of support for trusted others were mostly group-based and varied in content, focus, and mode of delivery. Organizational support included access to health care professionals [48,54,55], financial stipends [55], psychoeducational support [40,54], and resiliency and emotion management training [38,46,48,54].

The PCAFC program was the only established national support program for care-partners of veterans [55]. In their analysis of administrative data from the US Department of Veterans Affairs, Banigan et al. [55] sought to inform care-partner support policy. Using data from 942 veterans and their trusted others, the authors reported that increased use of the PCAFC program by trusted others resulted in a 35 percent increase in veteran-supported employment. The program also provided training, ongoing monitoring of the care-partner and veteran by a clinician, and short-term financial aid [55]. The Coming Home Project (CHP) hosted three therapeutic retreats for (a) veterans, (b) active service members, and (c) their family members [38]. Education and support interventions were provided in large group meetings, small peer-based support groups, and individual sessions [38]. The training was described as psychoeducational and included mindfulness, relaxation, parent education, couples’ communication, recreational activities, and information on area-specific further supports [38]. A total of 175 veterans, service members, and their families completed program evaluations and reported a significant reduction in stress and isolation and improvements in relaxation and hope [38]. Family members responded favorably to the program and requested specific couple group time to work on specific communication problems [38]. Two formal sources of resilience and emotional training included: (a) the FOCUS Family Resilience Program [48,54] and (b) the Warrior Resilience and Thriving (WRT) program [46]. The FOCUS Family Resilience Program was an eight-session intervention for US military families experiencing combat and deployment-related stress aimed at promoting family cohesion and resiliency using structured narrative communicative approaches [48,54]. Lester et al. [48] observed high program satisfaction and a significant reduction in psychological distress across parent–child, family, and global functioning. These improvement scores were associated with participation in structured peer support, where families were allowed to share their deployment-related experiences and bridge communication gaps [48]. Saltzman et al. [54] described this narrative-sharing exercise as a way to build perspective-taking skills and reduce misattribution and estrangement. Another resiliency-training program that included trusted others, the Warrior Resilience and Thriving (WRT) program, was a pilot support program for soldiers and their families prior to, during, and following combat deployment [46]. This eight-hour program promoted resiliency, emotional management, and critical thinking. Training content included rational emotive behavior therapy (REBT), self-coaching, stoic principles, and survivor and resiliency strategies. Qualitative program evaluation data was supportive, although it is unclear if any of the views shared are those of a trusted other. The number of trusted others undertaking the program and the effectiveness of this support from their perspective are not reported.

#### 3.3.2. Informal Coping and Use of Resources

We define informal coping and the use of resources as the natural efforts of trusted others to alleviate the effects of secondary stress without the help of a mental health professional or organization-based structural support. Seven of the included studies describe the wide range of tools and skills trusted others use to prepare them for their supportive role. We identified two sub-themes of informal coping and use of resources, intrapersonal (self-management) and interpersonal (peer and community management).

Friese [44] reported self-management coping strategies as the most common resource trusted others use. These include self-taught or self-regulated emotion management skills [37,39], empowering beliefs [36,37,44], and problem-focused strategies such as information gathering and action planning [40,42]. In three of the included studies, trusted others reported managing their feelings and reactions toward first responders’ stories as a crucial action [37,42,46]. Moreover, other studies highlighted the use of meditation and self-care activities [38], developing individual interests (e.g., gardening and reading) [12], and having independent friends outside the first responder’s community [12] as positive coping strategies. Bochantin [39] found humor to be a coping mechanism and an important tool to lighten the mood and alleviate tension for children in first responder families. McKeon et al. [49,50] established that physical activities improve trusted others’ mental health and prepared them to provide support for first responders. Four studies refer to the core or religious beliefs as a personal resource that trusted others draw upon in times of stress [36,37,44]. Belief systems were described as a commitment and persistence to confront challenges and share the first responders’ burden [37], a sense of responsibility for first responders’ wellbeing [37], being hopeful about first responders’ recovery [37], a sense of loyalty and pride [44]. Three studies refer specifically to religious beliefs and the use of prayers as a coping strategy [36,44,51]. In a study of coping activities of police officers and their spouses, religious belief and practice were positively associated with emotion and problem-focused coping and negatively associated with divorce and maladaptive coping strategies [36]. Religion is described as the stable ground on which trusted others stand while continuously providing support for first responders. Self-education through information gathering was a key finding in relation to the preparation for the role of support partner. Buchanan et al. [40] and Ewles [42] described information gathering as related to their practical preparation and anticipation of challenges, especially trusted others of first responders with PTSD. In a qualitative study exploring perspectives of PTS, female spouses of combat veterans reported learning about PTSD through informal sources [40]. They decided to learn about PTSD to understand better their first responders’ needs and ways to support them. Beehr et al. [36] reported trusted others developing an action plan where they evaluate challenges and establish actions to improve their condition.

Multiple studies [37,44,47] acknowledge peer support, behavior monitoring, and safeguarding skills for trusted others. Knowledge sharing and emotional support through peer-to-peer interaction were identified most frequently as the way trusted others prepare themselves for their support role and the most frequently cited coping strategy [42,47]. As one trusted other stated, “I have deepened my bonds with other law enforcement wives … I get personal and emotional relief from talking to [them]” [47] (p. 314). Peer community, whether online or in person, was identified as a valuable resource for trusted others seeking support and practical knowledge, especially for those who experienced isolation and loneliness [37]. Interacting and involving trusted others in communities and organizations that understand trusted others’ and first responders’ experience was crucial in supporting trusted others’ resiliency. Multiple studies mentioned trusted others’ need to connect with others with similar experiences [37,44,47] and difficulties relating to others who do not understand their hardship [55]. Two studies discussed behavior monitoring and safeguarding in interactions with first responders. In one study, family members of veterans with PTSD reported using caution and sensitivity when interacting with first responders to avoid conflict and triggering their first responder’s symptoms [37]. Ewles [42] found that many trusted others developed a sensitivity toward their first responder’s verbal and non-verbal cues when facing adverse events. Trusted others reported the need to protect their wellbeing by establishing boundaries and using assertive language to express how they expected to be treated when providing support [36]. Interestingly, the trusted others of frontline healthcare workers also experienced guilt over protecting their safety during the COVID-19 pandemic [56]. Three studies described the similarities of coping strategies and use of resources between trusted others and first responders [36,44,55]. Tentative evidence suggested that first responders and trusted others coping strategies tend to mirror rather than complement [36]. Using prayer or alcohol as a means of coping in one spouse was often mirrored in the coping strategy adopted by the other. Friese [44] concurred, suggesting that compartmentalization between personal feelings and reactions when providing support was mirrored between both groups.

### 3.4. Theme 3-Support Needed


*“I lived my life walking on eggshells and how I made the kids walk on eggshells.”*
[37] (p. 651)

Several trusted others [42,46] describe their role as unbelievably all-consuming, overwhelming, and lonely. Trusted others often encounter stigma [40,42], lack of accessible resources [42,55], and barriers [12,41,42] in their role. Figure 3 summarizes the support needs of trusted others. Buchanan et al. [40] reported that 20 out of 34 trusted others identified stigma as the reason their first responder did not seek treatment for PTSD. The career and social ramifications of admitting to a mental illness were explored in terms of this negative bias [42]. The stigma of seeking help and disclosing symptoms was exacerbated by the toxic masculinity informally promoted within the profession [40]. Two studies cited the social lack of understanding of the condition as a significant barrier to support [41,54]. Additionally, there were multiple logistical barriers to social support provision, such as trusted others’ time and capacity [42,55], limited access [54], and negative perception of professional care or support system [37,57]. Several factors created personal barriers to accessing and providing support for first responders: (a) trusted others’ lack of information and skill to manage symptoms of mental illness [40,41], (b) lack of acknowledgment from formal service providers and healthcare providers [37,57], and (c) lack of social service literacy [55]. Multiple studies identified the lack of communication skills [42] and maladaptive coping strategies and core beliefs [11,36] as significant barriers to social support. Ewles [42] documented the moderation effect of trusted others’ skills, available resources, and goals in the outcomes of support interaction, in which trusted others acknowledge their skill gap in communicating and providing effective social support. Maladaptive coping behavior included hyper-masculinity [11,36], avoidant coping [36,43], withdrawal [42,47,53], compartmentalizing [12,42,47], disempowering beliefs [40], substance abuse [42].

#### Recommendations

Trusted others described their first responders’ behavior in terms of its polarity along a continuum from adaptive to maladaptive, from their usual cheerful self to explosive rage, and from disciplined to helpless. Based on the literature, we identified practical, treatment, and social support needs that we present as support recommendations derived from the perspective of trusted others’ descriptions of what they feel would help them provide support, summarized in Table 5. Although not a focus point for this paper, our literature review highlighted a dearth of measurable social outcomes or benefits for family members or first responders following engagement in social support. We recommend that future research must address this lack of measurable evaluation of the outcomes of social support for all participants.

## 4. Discussion

Our findings offer a more fine-grained understanding of the social support provided by trusted others, their preparedness to engage with their first responders, and their needs in providing support. By using House’s [24] framework, our review builds on existing work and identifies a range of emotional, instrumental, appraisal, and information supports that trusted others provide. We also delineate trusted others’ formal and informal support structures, resource utilization, and barriers experienced in their role. Our summary of key recommendations also provides not only concrete practical implications but also a roadmap for future research.

Our narrative review and synthesis in this space add to a small but growing body of work. Multiple studies have previously reported various combinations of support utilization. For example, Beks [37] and Buchanan et al. [40] reported how partners of veterans living with PTSD provided a combination of emotional, appraisal, and informational support in creating an encouraging environment for first responder treatment. Consecutively, Ewles [42] and Friese [44] highlighted partners of law enforcement officers incorporating instrumental and appraisal support in initiating flexibility and empathy toward their first responders. However, emotional support remains the most dominant type of support provided [12,37,40,43,52,53], closely followed by instrumental [12,47,51], appraisal [11,40,45,51], and informational support [40,42]. A shared understanding of the first responders’ challenges enables trusted others to reframe the negative reactions experienced and includes them as an essential source of support.

Our work demonstrates how trusted others utilize and seek formal and informal support for themselves to sustain their wellbeing. Along these lines, Lester et al. [48] and Saltzman et al. [54] also described an example of formal organizational support for the trusted others of military personnel, while Landers et al. [47] identified trusted others’ needs to find informal support networks with those sharing similar experiences as them. Friese [44] and Cases and Benuto [12] indicated that sharing their experiences within these groups is a valuable resource, offering practical support, providing an informed knowledge base, and building resiliency and connectedness among the group’s members. Unfortunately, providing social support can have a negative impact on trusted others [36,44,49,50,52], and a lack of recognition and understanding of this impact from organizations and society can act as a barrier to providing this support [11,51,57]. Moreover, stigma and maladaptive coping strategies such as hyper-masculinity [11], withdrawal [57], and substance abuse [36,44] are significant barriers to the provision of social support by trusted others.

We see several important ways forward to improve the capacity of trusted others to act and feel empowered in their role as support partners for their first responders. First and foremost, society must recognize the value of their role, acknowledge the practical barriers they face, and help resolve these issues by designing initiatives to further prepare them for this role. There is a clear need for family-centered organizational policies and training interventions that consider the wider impact of the first responder role, particularly on those closest to them. We recommend that training and interventions are established early and meet changing needs over time. Ideally, such training will offer the opportunity for skill development, which will require more detailed needs analysis in local jurisdictions and engagement of key stakeholder groups to ensure that any interventions are designed with local context and cultures in mind. Formal interventions and training programs for trusted others should consider the challenges experienced across the lifespan of the role. For example, the needs of new recruits and those transitioning to retirement differ vastly, and so will the needs of their trusted others. Multiple, real-time, and planned periodic supports need to be considered as an investment in the retention of emergency service personnel. Crucially, trusted others must be mindful of the support they provide and invest in themselves to continue in their caregiving role. Encouraging self-care in a realistic and attainable way for a busy supporter is suggested as an essential lifestyle intervention to enhance overall wellbeing.

Our review also identifies significant gaps in the current literature. The first responder community often focuses on specific occupations, such as police officers [36,42] or veterans [37,42] with particular conditions (e.g., PTSD) [12] or specific critical incident exposure (e.g., 9/11) [52] without recognizing the pattern of support preferences and challenges facing first responders and their loved ones.

We found an extreme lack of attention toward trusted others, despite documentation of secondary trauma spillover [12,28] and an urgent need for support specifically targeted toward their community [21]. Cox et al. [25] and Sharp et al. [58] summarized the outcomes of trusted others’ experiences and the impact on their health and wellbeing. Our results align with Cox et al. [25] and Sharp et al. [58] in acknowledging the lack of support and recognition of trusted others’ experiences in their daily lives. Additionally, our study summarizes the type of social support provided, the process by which it occurs, and the preparedness of trusted others for this support partner role. We outline an overview of trusted others’ capacity that, to our knowledge, has not yet been undertaken. Moreover, we modified the definition of trusted others, adapted from Evans [11], and we have expanded the recognition of first responders’ identity to frontline healthcare professionals to promote inclusivity in our research methodology. Based on these insights, we offer key priorities for a research agenda into trusted other support:Factors that foster the development of a comprehensive and culturally competent understanding of all facets of trusted other social support related to specific needs and experiences.Optimal research design and evaluation of interventions, programs, and training for trusted others’.Strategies to co-develop informal and formal support systems for trusted others and their first responder.

Our narrative review has several limitations. We explored the literature for papers outlining how trusted others support their first responders, how prepared they are for this task, and how they can be best supported while caring for their loved ones. Although rigorous, our review is not exhaustive, and we may not have included all studies in this heterogeneous literature base. Unlike a systematic review, we purposely did not formally rate the quality of included studies, instead choosing relevant papers guided by our research questions. We concur with Cox et al. [25] and Sharp et al. [58] regarding the dearth of research on public service personnel families and acknowledge that most of our included studies originated in the USA, Canada, Australia, and the UK. This highlights a lack of focus on this topic in Europe, among other international regions, which may not be representative of all trusted others’ experiences.

## 5. Conclusions

Trusted others play a significant role in the lives of their first responders in helping mitigate the negative impacts of duty-related traumatic exposures. Trusted other support contributes to first responders’ wellbeing, although providing this support also taxes trusted others who need support themselves. This tension highlights the need for consideration of the vital role trusted others play in the career of a first responder. Training and interventions for trusted others should take the various forms of support into account and proactively address trusted others’ needs. Future research should focus on the design, implementation, and evaluation of educational offerings for trusted others.

## Figures and Tables

**Figure 1 ijerph-19-16492-f001:**
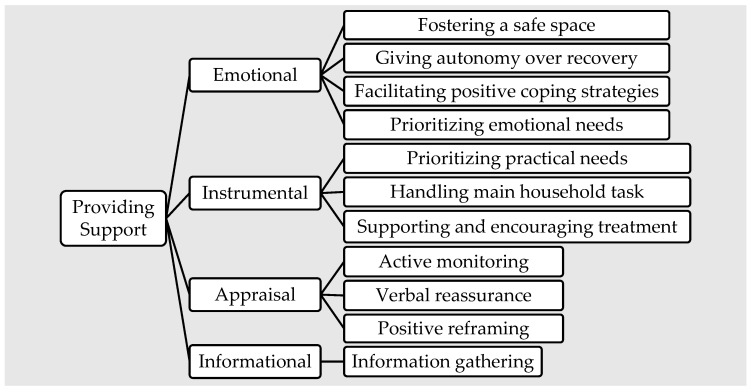
Provision of social support strategies categorized according to operational definitions (House 1981).

**Figure 2 ijerph-19-16492-f002:**
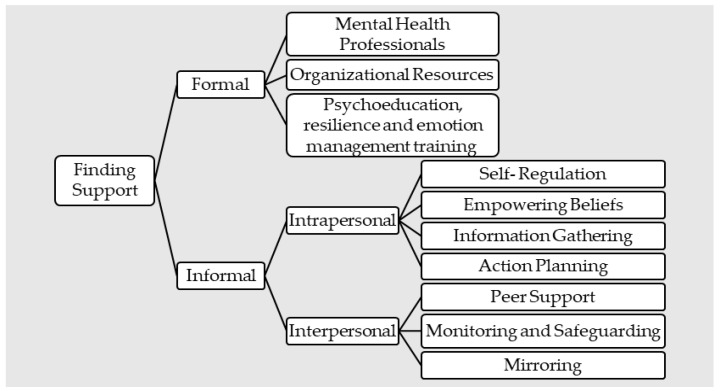
Summary of available support for trusted others.

**Figure 3 ijerph-19-16492-f003:**
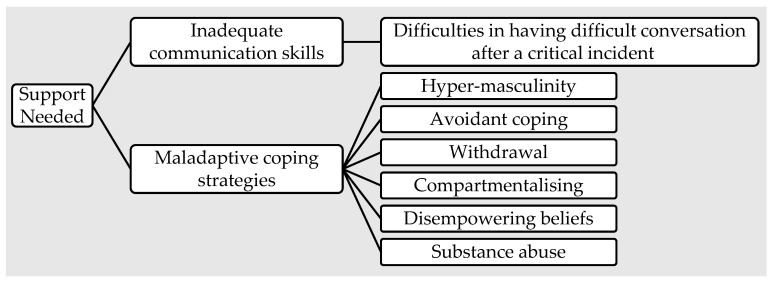
Trusted others’ support needs.

**Table 1 ijerph-19-16492-t001:** Inclusion and exclusion criteria.

	Inclusion Criteria	Exclusion Criteria
Participants	Trusted others (family member, partner, spouse, friend, or care-partner) of first responders.	Non-family member, partner, spouse, friend, or care-partner (e.g., clinician, counselor).
	First responders (military, veteran, law enforcement officer, fire fighter, paramedic or EMT, and frontline health care professional (physician and nurse practitioner)).	Non-first responders or frontline healthcare professionals.
Study Focus	Studies describing support strategies, coping mechanisms, training interventions, programs, or practices utilized by the trusted others of first responders.	Studies that do not acknowledge trusted others’ involvement in supporting first responders. Studies that do not describe trusted others’ social support provision, resource utilization, and barriers encountered in providing social support.
	Studies describing trusted others’ resource preference and availability (informal and formal), barriers, and limitations encountered in providing social support.	
Study Type	Qualitative, quantitative, or mixed methodologies, book chapters, thesis dissertations, and grey literature, including intervention protocols and manuals.	Literature reviews, review articles, reports, and conference papers were used to identify additional literature only.
Publication Characteristics	All countries, English language, full-text available, not restricted by publication date.	Non-English studies, no full-text available.

**Table 2 ijerph-19-16492-t002:** Summary of articles included in the narrative review.

Author (Year)	Study Design	Sample/Subject (Country)	Key Findings
Beehr et al. (1995) [36]	Quantitative	177 police officers and partners (USA)	Coping strategies: problem-focused, emotion-focused, religiosity, and rugged individualism. Strains previously noted in non-empirical literature were given special attention; divorce potential, drinking behavior, and suicidal thoughts.
Beks (2016) [37]	Qualitative	30 female partners of veterans (Canada)	Seeking out information and self-education to enhance understanding and aid healing. Reframing problems/needs from the individual to the collective (couple/family).
Bobrow et al. (2012) [38]	Quantitative	347 veterans, military service members, and family (USA)	Evaluation of “Coming Home Project” intervention. Peer groups enable shared experiences and mutual trust in a safe environment. Connect with a wider peer community and professional service providers.
Bochantin (2017) [39]	Qualitative	18 police officers, 18 firefighters, and 107 family members and spouses (USA)	Trusted others hide their fears for the safety of their first responders. Self-taught acceptance and self-regulation practice. Maintain family and personal equilibrium. Advantages and disadvantages of humor are identified.
Buchanan et al. (2011) [40]	Qualitative	34 spouses/partners of veterans (USA)	Two-thirds of the participants reported not having received formal education about PTSD. Barriers to seeking treatment were disclosure issues such as denial of symptoms, fear, and stigma. Partners were monitoring veterans for signs of PTSD symptoms or behaviors. Supporting their veteran partner to seek treatment.
Casas & Benuto (2022) [12]	Systematic review	904 FRs, 903 family members (USA)	Trusted others take on additional responsibilities to reduce demands on their first responder and flexibility around their spouse’s work schedule. Trusted others seek support from other family members and clergy, acceptance of what they cannot control. Trusted others cope by developing independent interests such as reading, gardening, and sewing.
Cyr et al. (2022) [41]	Mixed methods	16 veterans and partners (Canada)	Trusted others involved in treatment facilitation and have a protective support role in first responders’ recovery. Trusted others also reflected on the role they played in their partners’ recoveries. Need to align goals of treatment to support re-engagement in social functioning.
Evans et al. (2020) [11]	Qualitative	21 veterans and trusted others (USA)	Facilitating trusted others’ participation in first responders’ treatment. Variety of symptoms in real-world settings. Protective behaviors, e.g., actively watching for triggers. Informal care-partner without social or organizational acknowledgment. Escalating help-seeking when required.
Ewles (2019) [42]	Mixed methods	38 police officers and partners, and 179 FRs (Canada)	Communication skills utilized (e.g., paraphrasing what was said, having open-ended questions). Time out method (giving time and space to decompress). Reconnecting (scheduling quality family, one-to-one time). Linking to professional help.
Folwell & Kauer (2018) [43]	Qualitative	25 EMS (USA)	Providing time and empathy to their first responders. FRs using technology (e.g., apps, texts, or emails) as a barrier.
Friese (2020) [44]	Mixed methods	171 law enforcement officer spouses/partners (USA)	Self-care and exercise were the most used positive coping skills. Peer support, family time, and spirituality are also outlined as positive coping practices. Maladaptive practices included isolation and the use of alcohol.
Hill et al. (2020) [45]	Qualitative	10 family members of the fire and rescue services personnel (UK)	Accommodating and sacrificing: prioritization of the first responders meant family life was built around the needs of the organization. Monitoring reaction after critical incidents, with experience, they became accustomed to “reading” their first responders. Facilitated known coping strategies: through further conversations to diffuse the firefighter or other emotional and practical ways of coping.
Jarrett (2013) [46]	Mixed-methods: Program evaluation	Soldiers and family members (USA)	Soldier and family cognitive resiliency training classes. Therapeutic approach based on Rational Emotive Behavior Therapy.
Landers et al. (2020) [47]	Qualitative	8 spouses of law enforcement officers (Canada)	Practical help (e.g., household duties and roles, caregiving). Fostering a safe space for the first responders. Fostering a safe space for the first responders. Providing encouragement, empathy, and humor. Resilience from remembering the good times.
Lester et al. (2012) [48]	Quantitative	331 military families (USA)	Increasing family cohesion: sharing experience and understanding, bridging communication gaps. FOCUS family resiliency training: 30 to 90 min sessions, standardized and manualized. Core FOCUS skills tailored to address specific family needs.
McKeon et al. (2021) [49]	Quantitative	34 EMS workers and 30 partners (Australia)	High rates of psychological and physical morbidity among support partners require intervention and treatment including physical and mental health lifestyle interventions.
McKeon et al. (2022) [50]	Quantitative	47 EMS workers and 43 partners (Australia)	Physical activity as a coping strategy. Social media as a mode of delivery. Mental health informed physical activity program.
Menendez et al. (2006) [51]	Qualitative	26 spouses of firefighters (USA)	Identified the need for monitoring and protective behavior, e.g., seeking formal support, utilizing connectedness, and healing. Lack of recognition and formal organizational support.
Porter and Henriksen (2016) [52]	Qualitative	6 spouses of first responders (USA)	Being the ‘‘sounding board’’ by providing positive encouragement, constant communication, and verbal reassurance. Identified the “burden of care.”
Roth & Moore (2009) [53]	Qualitative	14 spouses and parents of EMS workers (USA)	Emotional support; being there, giving space and autonomy. Disassociation following stressful events.
Saltzman et al. (2016) [54]	Program description	Case examples of trusted others dealing with trauma (USA)	FOCUS family resilience program: a family-centered strength-based program. Selected preventive services to families that build perspective-taking skills and mutual understanding (narrative-sharing techniques). Reducing distortions and misattributions: mitigating estrangement in the family.
Shepherd-Banigan et al. (2022) [55]	Quantitative	Secondary analysis of administrative US veteran affairs data (USA)	National program of comprehensive, enhanced support for family care-partners of veterans. Mandatory caregiving training, ongoing monitoring of the veteran and trusted others by a clinician (home/phone-based assessments). Financial stipends to alleviate short-term financial burdens. Allowing trusted others to be present to support first responders.
Tekin et al. (2022) [56]	Qualitative	14 family members of frontline healthcare professionals (UK)	Consideration of the wide-reaching impact trusted others, e.g., emotional and physical burden, increased roles, responsibilities, and lack of self-care. Instrumental supports; family-friendly policies and practices, personal medical dictionary, adequate testing, long-term medical follow-up, and support. Mitigating factors: pride in the job, sense of security derived from first responders’ medical knowledge, surviving the job.
Waddell et al. (2020) [57]	Qualitative	10 partners of veterans (Australia)	Identified needs for social support due to lack of understanding from “outsiders.” Partners were defined in healthcare settings as support persons with no acknowledgment of their relationship to veterans. Identified the importance of advocating for the partner and help-seeking.

EMS: Emergency medical services; FOCUS: Families Over Coming Under Stress; FRs: First Responders; PTSD: Post-traumatic stress disorder.

**Table 3 ijerph-19-16492-t003:** Summary of findings.

Main Themes	Sub-Themes
Providing Support	
Emotional support	Fostering a safe space: being present, a sounding board, and expressing acceptance.
	Giving autonomy over first responders’ recovery.
	Facilitating first responders’ positive coping strategies: seeking professional help, practicing religious activities, and constructive conversations.
	Prioritizing first responders’ emotional needs over the trusted others’ personal feelings.
Instrumental support	Prioritizing first responders’ practical needs.
	Handling main household tasks and taking on additional responsibilities.
	Supporting and encouraging treatment: arranging healthcare appointments and advocating for first responders in a healthcare setting.
Appraisal support	Actively monitor first responders’ condition and express concern and feedback.
	Verbal reassurance over first responders’ capability and motive in working as a first responder.
	Reframing difficulties in a positive light and providing positive feedback.
Informational support	Gathering information on first responders’ treatment plans, symptoms, and triggers.
Finding Support	
Formal	Mental Health Care Professionals.
	Organizational resources: access to healthcare, financial stipend, psychoeducational support, and resiliency and emotion management training.
Informal	Intrapersonal (self-management): self-regulation skills, empowering beliefs, problem-focused strategies (information gathering and action planning).
	Interpersonal (peer and community management): peer support, behavior monitoring, and mirroring.
Support needed	
Inadequate communication skills	Difficulties in having difficult conversations with first responders after a critical incident.
Maladaptive coping strategies	Hyper-masculinity, avoidant coping, withdrawal, compartmentalizing, disempowering beliefs, substance abuse.

**Table 4 ijerph-19-16492-t004:** Definition and categorization of House (1981) social support framework.

Category	Definition
Emotional support	The provision of trust, empathy, love, and care
Instrumental support	The provision of tangible services, goods, or aid
Appraisal support	The provision of affirmation or communication of information that is relevant for self-evaluation
Informational support	The provision of knowledge and information

**Table 5 ijerph-19-16492-t005:** Support recommendation.

Area	Actions
Practical support [41,49,54,56]	Assist with childcare and finances
Assessment and treatment [11,38,39,41,43,50]	Involve trusted others in assessment and treatment Align treatment goals with those identified by family and first responders Focus treatment on social functioning and emotional literacy Offer timely and adequate assessment Identify trauma-informed care, interventions, and facilitators
Organizational support [39,40,41,45,50,51,52,53,55,56]	Establish family-friendly policies and practices Provide organization-based training to include consideration of the wide-reaching impact on family Design formal intervention programs for trusted others
Information and training [38,40,41,43,45,47,48,49,51,52,53]	Offer communication skills training (e.g., difficult conversations after a critical incident, conflict de-escalation, relationship fostering) Provide early interventions, information, and resource provision Need for experiential learning environments, supports, and theory as reading in the text does not capture reality
Social support [41,42,45,48,49,52,57]	Facilitate culturally competent peer-support networks and communities Provide adaptable, continuous support over the span of first responders’ career duration (needs change over time) Offer accessible and flexible modes of support and treatment delivery (e.g., online communities)
Lifestyle intervention [37,45,46,52]	Support the physical and mental health of trusted others due to the effect of stress and lack of self-care

## Data Availability

Not applicable.

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
