# Peer review of "Turning to ‘Trusted Others’: A Narrative Review of Providing Social Support to First Responders"

_ijerph, 2022, doi:10.3390/ijerph192416492_

Round 1
Reviewer 1 Report
This is a review of means and varieties of support provided by “trusted others” (friends and family) documented in the literature for first responders, as opposed to professional-level/institutional-provided support or therapies. The introduction frames the evaluation in terms of PTSD outcomes. The purpose was to provide or categorize the types of support offered by trusted others documented in the literature.
To that purpose, it succeeded. The strengths of this “narrative review” are the rigor of the literature search, the involved nature of the authors, and the ability of the authors to summarize a wide-range of research/report designs. The review reads like a well-written textbook article with appropriate definitions and organization.
A distinct weakness of this article lay in the absence of outcomes; to me, PTSD and social outcomes are the underlying goals that fuel this research. Whatever the discovered articles uncovered, and whether the cited study was “qualitative” or “quantitative”, no attempt was evident in linking the various types of “trusted other” support to any measure of effectiveness in their use. I was left with the following questions after reading this:
-what are the outcomes of PTSD severity/remission/suicide/job retention linked with any facet of “trusted other” support?
-what are the outcomes of the networks of trusted supporters when faced with a first responder with PTSD?
-what are outcomes of those first responders who reject professionally available therapies compared to those who rely solely on trusted support networks?
Perhaps the overt purpose was to provide a type of organizational tree as to how non-professional networks support those vulnerable to PTSD, but I was left with the impression that the outcome part was distinctly missing. If these questions are those that the authors want to answer, and the literature is lacking, then some clear cut suggestions on where literature should go are needed. The authors in Discussion “Factors that foster the development of a comprehensive…” alluded to needed research, but framing that quest in clear, outcome related goals would be preferred to the more cloudy goals provided in the text. If, on the other hand, some of these questions are available in the cited research, then some information on outcomes would be highly useful to provide clinical importance to the attention to the reader. At the very least, the mannuscript should tell us why outcomes cannot be linked to the different strategies summarized.
Reviewer 2 Report
Thank you very much for having the opportunity to review an interesting article, the aim of which is to mataanalyze publications in the scope of social support obtained by first aid workers. The problem area raised is not only of theoretical, but also of applied importance. The article is very carefully prepared from the merits - clearly formulated purpose and research questions, presented results and discussion. From a methodological point of view, the work does not bring any objections. The methods of data analysis used in the work are well described, appropriately selected. The conclusion is correct and based on the methods described in the previous sections. The description of the results is clear and easy to understand.
Author Response
Dear Reviewer,
We thank you for comments. Through this publication, we hope to acknowledge the importance of informal support trusted others have been providing and bridge the understanding of support behaviours, coping strategies, and resources this community needs.
We thank you once again for your comments and hope this revision meets your approval.
Warm Regards,
The authors
Round 2
Reviewer 1 Report
I have no further comments